# Differential Cytokine Responses and the Clinical Severity of Adult and Pediatric Nephropathia Epidemica

**DOI:** 10.3390/ijms24087016

**Published:** 2023-04-10

**Authors:** Ekaterina Martynova, Robert J. Stott-Marshall, Venera Shakirova, Albina Saubanova, Asiya Bulatova, Yuriy N. Davidyuk, Emmanuel Kabwe, Maria Markelova, Ilseyar Khaertynova, Toshana L. Foster, Svetlana Khaiboullina

**Affiliations:** 1“Gene and Cell Technologies” Institute of Fundamental Medicine and Biology, Kazan Federal University, OpenLab, 420008 Kazan, Russia; 2Faculty of Medicine and Health Sciences, School of Veterinary Medicine and Science, The University of Nottingham, Sutton Bonington Campus, Loughborough LE12 5RD, UK; 3State Medical Academy, 420012 Kazan, Russia

**Keywords:** nephropathia epidemica, cytokine, pediatric

## Abstract

Nephropathia epidemica (NE), caused by the hantavirus infection, is endemic in Tatarstan Russia. The majority of patients are adults, with infection rarely diagnosed in children. This limited number of pediatric NE cases means there is an inadequate understanding of disease pathogenesis in this age category. Here, we have analyzed clinical and laboratory data in adults and children with NE to establish whether and how the disease severity differs between the two age groups. Serum cytokines were analyzed in samples collected from 11 children and 129 adult NE patients during an outbreak in 2019. A kidney toxicity panel was also used to analyze urine samples from these patients. Additionally, serum and urine samples were analyzed from 11 control children and 26 control adults. Analysis of clinical and laboratory data revealed that NE was milder in children than in adults. A variation in serum cytokine activation could explain the differences in clinical presentation. Cytokines associated with activation of Th1 lymphocytes were prominent in adults, while they were obscured in sera from pediatric NE patients. In addition, a prolonged activation of kidney injury markers was found in adults with NE, whilst only a short-lasting activation of these markers was observed in children with NE. These findings support previous observations of age differences in NE severity, which should be considered when diagnosing the disease in children.

## 1. Introduction

Nephropathia epidemica (NE), a mild form of hemorrhagic fever with renal syndrome (HFRS), is endemic in the Volga Region of Russia, with the Republic of Tatarstan (RT) among the most active foci [1,2]. Infection is characterized by an acute onset fever, malaise and headache [3]. These non-specific prodromal symptoms last from 3 to 4 days and are followed by specific symptoms of *orthohantavirus* infection [4]. These symptoms are mainly attributed to various degrees of acute kidney injury (AKI) which develops in up to 95% of NE patients [5,6]. The most common criteria for the diagnosis of AKI are decreased urine output, increased serum creatinine (sCr) levels and decreased creatinine clearance [7]. Signs of an increased hematocrit and mild mucosal bleeding, indicating vascular permeability and disturbed hemostasis, are described in the majority of patients [8]. Disease symptoms resolve within 14 days and complete recovery is expected [9,10]. However, in a substantial number of patients, the recovery may take 6 months or longer [11], likely explained by chronic renal damage [12].The major cohort of patients consist of adult males with a history of outdoor activity before diagnosis [13]. Pediatric NE is rare, with cases usually linked to outbreaks in families [14]. Additionally, a systemic literature review by Huttunen et al. supports a higher frequency of NE in adults when compared to children [15]. Authors analyzed data from 1968 to 2008, where 573 adult and 80 pediatric NE cases were diagnosed (1:6.7 ratio). Similarly, Makary et al. reported fewer numbers of pediatric cases when compared to adult NE cases in Finland from 1995 to 2008 [16]. In this study, out of 22,681 cases, only 1212 were pediatric NE cases (1:17.7 ratio). Symptoms of pediatric NE resemble those in adult patients [17]; however, data on the severity of symptoms and the frequency of AKI vary. A lesser severity of NE in children compared to adults was reported by Mustonen et al. [17]. However, another study by Echterdiek et al. reported the clinical course and frequency of AKI to be similar between the two age groups [18]. These differences may be explained by the number of patients in each study, as well as the potential variation in the virus strain involved. The sporadic nature of infection, however, as well as the limited number of patients identified, greatly hinders our current understanding of NE pathogenesis in children.

There are two phases of AKI, namely oliguric and polyuric [19]. The oliguric phase is characterized by decreased urine output, which in some patients could develop into transient anuria, a lack of urine output. The duration of this phase can be between 3 and 7 days. The recovery of renal function is indicated by polyuria with diuresis, with increases of up to several liters per day. Pathogenesis of AKI symptoms can be explained by the histological finding of tubulointerstitial nephritis [20,21]. Orthohantaviruses can infect tubular epithelial cells and podocytes [22], although infection appears to be noncytopathic, and the reduced barrier function of infected cells was suggested to explain proteinuria [23,24,25]. Multiple markers, such as hematuria, proteinuria, glucosuria, crystallin C and α-1-microglobulin were suggested for diagnosis and prognosis of AKI [26,27,28,29,30]. We, too, have demonstrated an increased level of urinary clusterin in NE as an indicator of kidney injury [31]. Absent in normal kidney function, clusterin is released exclusively by kidney epithelial cells [32], and thus could serve as a marker of tubular cell injury [33]. Another marker, IL-18, was found in NE patient urine, implying the presence of inflammation in the kidney parenchyma [31]. These biomarkers were identified in the urine of adult NE patients, though our knowledge of the clinical values of these markers in children diagnosed with NE remains insufficient.

In this study, we sought to determine whether the severity of NE differs in children compared to adult patients. We show that signs and symptoms of pediatric NE appear to be milder than in adults, and that clinical symptoms and signs of AKI were also significantly less pronounced in children. Fewer cytokines were activated in sera from pediatric patients, suggesting that the “cytokine storm” contributes more to disease pathogenesis in older patients. Furthermore, heightened activation of cytokines associated with activation of Th1 lymphocytes, and a prolonged activation of kidney injury markers was observed in adult patients, whilst they were only briefly activated in children.

## 2. Results

### 2.1. NE Serum Samples Collection Sites

Children and adult NE serum samples were collected during an *orthohantavirus* infection outbreak in 2019 in the Republic of Tatarstan (RT), Russia. The sites of *orthohantavirus* exposure are summarized in Figure 1. These locations are known to be endemic for orthohantavirus infection, and they were previously identified as habitats of bank vole populations carrying PUUV as well as sites of human infection [34,35].

### 2.2. Sequencing and Phylogenetic Analysis of PUUV Strains from Human and Rodents

In total, 140 serum samples (129 adults and 11 children; 1:11.7 ratio) were collected from patients diagnosed with NE during an *orthohantavirus* infection outbreak in 2019 in the RT. PCR analysis identified PUUV strains in 59 (45.7%) adults and 5 (45.5%) children NE. The PCR products of complete or partial S segment coding sequences (CDS) of 41 PUUV strains from NE patients were obtained. Partial S segment CDS sequences 590 bp long were used to generate the phylogenetic tree (Figure 2). All PUUV nucleotide sequences formed a clade with previously identified RUS PUUV genetic lineage [2,35]. Analysis of the *orthohantavirus* strains from NE patients and the PUUV strains previously isolated from bank voles revealed close relationships. In most cases, the PUUV sequence from rodents had a high similarity with sequences obtained from patients from the same location. This statement is confirmed by the analysis of PUUV strains Hu466, Hu475, Hu493, Hu497, Hu500, Hu505, Hu518, Hu520, Hu523, Hu546, Hu563, Hu566, Hu587, Hu599, Hu608, HU_611 and Hu614 from NE patients, that together grouped with strains from bank voles in the northwest regions of the RT (Figure 1, the green circles and names, and Figure 2, subclade Pre-Kama, North). In addition, Hu526, Hu545, Hu574, Hu 600 and Hu604 strains were closely related to PUUV strains from habitats of bank voles in the east and southeast suburbs of Kazan (Figure 1, brown circles, Pre-Kama, South). Additionally, Hu464, Hu549, Hu598 and HU_639 strains were closely related to PUUV strains from Mamadysh in the east of the Pre-Kama area (Figure 1, blue circles and names), and “Pre-Kama, Mamadysh”. The strains highlighted in purple on the phylogenetic tree and trapping map are clustered with strains from southwest of the Pre-Kama area and strains from the Western Trans-Kama area. It should be noted that Hu461, Hu471, Hu487, Hu488, Hu542, Hu578, Hu602, Hu603, Hu638 and Hu624 strains from NE patients and strains circulating in the bank voles from the forest near Teteevo in the Pre-Kama area had the highest node support on the phylogenic tree (Figure 1 and Figure 2). PUUV HU_585, HU_626 and HU_627 strains identified in pediatric NE formed one branch with bank vole PUUV strains from the Chistopol area. Additionally, one strain, Hu458, isolated from adult NE, was located in the branch with strains from the Starye Salmany village in the Western Trans-Kama area (Figure 1 and Figure 2). Finally, the Hu465 strain was grouped with strains from the Pre-Volga area in subclade Pre-Volga (Figure 1 and Figure 2, the red circles and names).

### 2.3. NE Clinical Laboratory Characterization

#### 2.3.1. NE Symptoms and Laboratory Data in Adults

Clinical data on adult NE signs and symptoms are summarized in Table 1. Of the 129 cases monitored, the majority of NE cases were classified as moderate (85.27%), while there were severe (10.07%) and mild (4.65%) forms of NE as well. All patients had fever (6.26 ± 2.45 days), and 37 patients (28.68%) had the second wave of fever lasting 3.5 ± 2.9 days. The majority of adult NE patients reported a headache (79.84%). Kidney injury symptoms were the most reported. These were pains reported in the lumbar region (85.27%), oliguria (71.31%) and anuria (3.88%). There were 60 patients diagnosed with AKI (46.51%), characterized by increased serum sCr (≥26.5 μM/L) for 48 h or decreased diuresis (<0.5 mL/kg/hour) for 6 h [36].

AKI stage I was diagnosed in 23 (17.82%) adult patients, which was characterized by sCr ≥ 26.5 μM/L during a 48-h period. Stage II AKI was diagnosed in 23 (17.82%) adults who presented with a 2–2.9-fold increase in sCr for 7 days when compared to typical levels. Additionally, 14 (10.85%) patients had stage III AKI, which was characterized by 3-fold higher sCr than typical levels for 7 days.

The second most common symptoms were those indicating irritation of the gastro-intestinal tract (GIT), such as abdominal pain (37.98%), nausea (41.09%), vomiting (29.46%) and diarrhea (33.33%). Patients also reported fog in the eyes (37.98%) and cough (14.73%). There were also reports of various types of hemorrhages (6.98%), which were characterized by nasal bleeding and scleral petechia (Table 1).

Laboratory results, summarized in Table 2, also demonstrated AKI in adults where serum levels of BUN and sCr were significantly increased 2–2.4-fold at the febrile phase and oliguric phases. BUN and sCr levels were lower in the polyuric phase but remained significantly higher than controls. In addition, serum levels of potassium ions were significantly elevated in the febrile and oliguric phases compared to controls. There was evidence of disturbed coagulation in adult NE patients, as laboratory tests revealed significantly lower thrombocyte counts in the febrile and oliguric phases. However, thrombocyte counts in the polyuric phase did not differ from controls (Table 2).

#### 2.3.2. NE Symptoms and Laboratory Data in Children

NE signs and symptoms in children are summarized in Table 3. Ten children had moderate NE (90.9%), while only one patient had severe NE. All patients had fever (6.64 ± 3.5 days); however, none experienced the second wave of fever. Only three patients (27.27%) had headache. Symptoms of kidney injury were commonly found in children. These were lumbar pain (36.36%), oliguria (63.64%) and AKI (36.36%). AKI stage I was diagnosed in one patient (9.09%), while stage II was found in two patients (18.18%). Additionally, AKI stage III was diagnosed in one patient (9.09%). None of the children with NE had anuria. Symptoms of the irritated GIT such as abdominal pain (45.45%), nausea (54.55%), vomiting (45.45%) and diarrhea (9.09%) were also reported. Fog in the eyes was found in three patients (27.27%), while none of the child NE patients reported having a cough. Hemorrhages were diagnosed in one patient and characterized by nasal bleeding.

Kidney injury signs were present in laboratory test results (Table 4). Pediatric NE patients had a significant 1.7–3.4-fold increase in serum levels of BUN at the febrile and oliguric phases. BUN serum levels also remained significantly increased during the polyuric phase. In addition, sCr were not significantly increased at the febrile phase, but were significantly higher during the oliguric and the polyuric phases. In these pediatric cases, we did not find changes in the serum levels of potassium ions in any phase. Signs of disturbed coagulation in pediatric NE cases were evident, as thrombocyte counts were significantly lower than controls in the febrile and oliguric phases. Thrombocyte counts did not differ from controls in the polyuria phase (Table 4).

#### 2.3.3. Clinical Comparison of NE in Adult and Pediatric NE Cases

Clinical and laboratory data showed differences which could be attributed to the age of the patient (Figure 3, Appendix A). NE patients from both age groups had fever of a similar duration (6.26 ± 2.45 days in adults vs. 6.64 ± 3.5 days in children, *p* = 0.83); however, the second wave of fever was not observed in children, while it was present in 28.68% of adults. In addition, more adults had headache as a symptom (79.84%) than children (27.27%), and the duration of headache was longer in adults compared to children (4.52 ± 2.91 days vs. 1.27 ± 2.57 days, *p* = 0.001). The duration of hemorrhagic syndrome did not differ between these patient age groups; however, the prolonged duration of 10 days was only observed in adults. Hemorrhagic syndrome was found in nine (6.98%) adults and only in one (9.09%) child diagnosed with NE. Cough was not documented in children, while it was reported in 19 (14.73%) adults with NE. In one adult the cough lasted for 15 days. The GIT irritation symptoms, nausea, vomiting and abdominal pain were found in both groups of patients. The duration of lumbar pain symptoms was longer in adults when compared to pediatric NE patients. Symptoms of kidney injury were present in both age groups of NE patients; however, these symptoms were more pronounced in adults. More adults had lumbar pain symptoms (85.27% vs. 36.36%). In addition, decreased diuresis was longer in adults compared to children with NE. Additionally, 5 (3.88%) adults experienced anuria, while it was absent in children. Having increased kidney injury as a result of NE in adults is indicated by significantly higher sCr in adults in the febrile and polyuric phases compared to children. These data indicate that NE symptoms tend to be more severe in adults than in children.

#### 2.3.4. Analysis of Serum Cytokines and Urine Kidney Damage Markers in Adult and Pediatric NE Cases

The dynamics of serum cytokines and urine kidney damage markers were analyzed in adults and children with NE. Data were analyzed during three phases: febrile (1–7 days), oliguric (8–11 days) and polyuric (>12 days) (Appendix A). In the febrile phase, 28 cytokines (IL-1β, IL-2, IL-2Ra, IL-3, IL-4, IL-5, IL-7, IL-9, IL-10, IL-12p40, IL-15, IL-16, IL-17, IL-18, LIF, M-CSF, CCL2, CXCL9, CCL3, CCL4, CCL5, CCL27, CXCL1, GM-CSF, IFN-α2, TNF-α, TRAIL and VEGF) were increased in adult NE patient serum (Figure 4). We also found changes in kidney toxicity markers, i.e., levels of clusterin were increased, while calbindin were decreased. In contrast, levels of only 12 cytokines (IL-1Ra, IL-1α, IL-2Ra, IL-4, IL-10, CXCL10, CCL11, CCL2, CCL27, MIF, SDF-1a and TNF-α) were increased, while IL-8 levels were decreased in pediatric NE patients. Levels of clusterin were increased in children, similar to adult NE patients. However, in contrast to adults, pediatric NE patients showed an increase in calbindin levels.

When cytokines and kidney toxicity markers were analyzed in the oliguric phase, serum levels of 29 cytokines (IL-1β, IL-2, IL-2Ra, IL-3, IL-4, IL-5, IL-7, IL-9, IL-10, IL-12p40, IL-15, IL-16, IL-17, IL-18, CCL2, CXCL9, CCL3, CCL4, CCL5, CCL27, CXCL10, GM-CSF, CXCL1, LIF, M-CSF, IFN-α2, TNF-a, TRAIL and VEGF) remained increased in adult NE patients (Figure 5), including all cytokines that were upregulated in the febrile phase with the addition of CXCL10 and HGF. However, in children, fewer cytokines remained upregulated (9 cytokines: IL-1α, IL-2Ra, IL-4, CCL2, CCL4, CCL11, CCL27, CXCL10 and TNF-α). The kidney toxicity marker clusterin remained increased in adult urine, similar to the febrile phase; in pediatric NE patients, levels of both clusterin and calbindin remained increased (Figure 5).

In the polyuric phase, a substantial number of analytes (30 cytokines; IL-1β, IL-2, IL-2Ra, IL-3, IL-4, IL-5, IL-7, IL-9, IL-10, IL-12p40, IL-12p70, IL-15, IL-16, IL-17, IL-18, CCL2, CCL3, CCL4, CCL5, CCL27, CXCL1, CXCL9, CXCL10, IFN-α2, LIF, GM-CSF, M-CSF, TNF-α, TRAIL and VEGF) remained upregulated in adult NE patient serum (Figure 6). Urinary levels of kidney toxicity markers remained affected similarly to that in febrile and oliguric phases. In child NE patients, similarly to the febrile and oliguric phases, the number of cytokines affected (IL-1Ra, IL-1α, IL-2Ra, IL-4, IL-10, CCL2, CCL11, CCL27, SCF, CXCL12, TNF-α and IL-8) was less when compared to adult NE patients (11 vs. 30 cytokines, respectively). Additionally, in the polyuric phase, children had higher levels of urinary calbindin than controls. In contrast, urine levels of clusterin were not different from controls (Figure 6).

## 3. Discussion

We have shown a lesser severity of NE in children when compared with adults during an outbreak in 2019 in RT. Amongst the clinical signs and symptoms defining NE, children had a shorter duration of oliguria and fewer had lumbar pain than adults. In addition, none of the children had anuria, while this was documented in 6% of adults. Clinical laboratory data indicated a lower intensity of kidney injury in children than adults. This was indicated by lower sCr and BUN. These differences in NE severity could not be explained by the variation of orthohantavirus strains isolated from children and adults, as only PUUV strains were found in both age groups. It should be noted that even the orthohantavirus lineages identified in children had a substantial similarity with those isolated from adult NE patients. These data suggest that the milder form of NE in children than in adults was most likely the result of age-related differences in the immunological responses to infection that can influence disease progression and clinical manifestation. Our data concurs with the previous observation made by Mustonen et al., that NE has lesser severity in children compared to that in adults [17]. In our study, clinical differences were corroborated by age-related variations in serum cytokine and urine kidney toxicity markers. These markers could be used to advance our understanding of NE pathogenesis in adults and children.

The differences in age-related NE severity are also supported by our finding of fewer serum cytokines affected in children than in adults; fewer cytokines were affected in children in all phases of the disease. These data indicate that the host reaction to infection in children has reduced intensity compared to that in adults. Therefore, we surmise that the early cytokine activation, or cytokine storm, contributes to pathogenesis of NE in adults but has lesser significance in children. Amongst the cytokines elevated in serum of adult NE patients, many have pro-inflammatory properties: IL-1β, IL-18 and TNF-α. Interestingly, IL-1β and IL-18 appear to be elevated only in adults, while serum levels of these cytokines remained unaffected in children. These cytokines are products of activated inflammasomes [37] serving as mediators of inflammation. Additionally, these cytokines, in combination with others, such as CCL5, CXCL9 and CXCL10, contribute to shaping the immune response [38] IL-1β produced by antigen presenting cells (APCs) is shown to support T cell priming [39], while IL-18 can support the commitment of Th1 lymphocytes [40,41]. Additionally, Th1 lymphocyte differentiation can happen when IL-18 and IL-12 induce the production of IFNγ [42]. Our data supports activation of Th1 lymphocytes in adult NE patients, with high serum levels of IL-1β, IL-18 and IL-12. Interestingly, high serum levels of these cytokines were found in febrile, oliguric and polyuric phases, indicating extended activation of Th1 lymphocytes in adult NE patients.

Along with Th1, evidence of the early activation of the Th2 immune response was found in adult NE patients. We detected high serum levels of IL-4 and IL-10. This Th2 lymphocyte activation was found in the oliguric and polyuric phases. Th1 immune responses appear to be protective, as the anti-orthohantavirus glycoprotein-specific CD4+T-cell response was shown to inversely correlate with the virus load in patient plasma [43]. It was also demonstrated that the mild form of infection develops in patients with a broad antigenic repertoire of Th1 cells. Additionally, the cytotoxic activity of orthohantavirus-specific cytotoxic T cells (CTCs) was demonstrated in vitro [44,45]; it is likely that by participating in the virus clearance, these CTCs could also contribute to the severity of the disease. This assumption is based on data published by Kilpatrick *et al.*, where a higher frequency of orthohantavirus epitope-specific CTCs was found in patients with the severe form of the disease [46]. Higher serum levels of cytokines activating Th1 lymphocytes suggests an early and prolonged activation of the Th1 immune response, which could contribute to enhanced disease severity in adults.

Interestingly, activation of the Th2 immune responses is evident in pediatric NE patients, as high serum levels of IL4 and IL-10 were demonstrated in febrile, oliguric and polyuric phases. Th2 lymphocytes assist the differentiation of B cells and production of antibodies. Antibody responses have been detected in orthohantavirus-infected patients, as high levels of orthohantavirus IgM antibodies were detected at the early stages of the disease [47]. Additionally, early activation of neutralizing antibodies has been demonstrated in orthohantavirus patients [48]. These neutralizing antibodies could have a protective role, as lower titers were shown in patients with severe forms of orthohantavirus infection [49]. Our data on serum cytokine levels demonstrate activation of Th2 cytokines in children, while Th1 cytokines were found increased in the serum of adult NE patients. These variations in cytokine activation suggest a potential contribution of Th2 immune responses to the comparatively lower severity of NE in children.

We also report substantial differences in the serum levels of kidney injury markers, clusterin and calbindin. Serum levels of clusterin were elevated in both age groups only in the febrile phase. However, it was decreased in the oliguric phase and unaffected in the polyuric phase in children, while it remained increased in adults. Increased clusterin expression has been demonstrated in injured kidneys, while absent in normal kidney tissues [32,50]. Expression of clusterin is found in the tubular cells of injured kidneys [33,51]; it is suggested that the secreted form of clusterin is protective against oxidative stress [52,53,54] and injury due to deposited immune complexes [55]. The protective role of clusterin was also demonstrated by Nguan et al., who showed that it is required for renal tissue regeneration by promoting tubular tissue proliferation [56]. Our data suggest that kidney injury is prolonged in adults compared to children.

Pediatric NE was characterized by persistently increased calbindin levels, which was decreased in adults. Calbindin is a calcium binding protein located in the distal convoluted tubule (DCT) [57]. Calbindin release could be triggered by low levels of Ca2+ as well as low levels of active vitamin D metabolite 1.25-(OH)2D [57]. Interestingly, low serum levels of Ca^2+^ has been demonstrated in adult NE patients [58]. In addition, decreased serum levels of vitamin D was reported in adult NE patients [58]. In addition to low serum levels of Ca2+, hyperparathyroidism was reported in orthohantavirus-infected patients [58]. Thus, our data contributes to the understanding of the mechanisms of Ca^2+^ homeostasis in adult NE patients. We believe that low calbindin levels could contribute to decreased Ca^2+^ reabsorption in DCT, which could explain the increased compensatory parathyroid gland reactivity. Our findings in children suggest the limited effect on Ca^2+^ reabsorption in the kidney, as levels of calbindin remained increased in this group of NE patients.

In conclusion, we have demonstrated substantial differences in clinical and laboratory data in pediatric and adult NE patients. Our data provide evidence of a milder form of NE in children than adults. We suggest that variations in cytokine activation could explain the differences in clinical presentation. Cytokines associated with activation of Th1 lymphocytes are prominent in adults, whilst they were absent in pediatric NE patient serum. Additionally, long-lasting activation of kidney injury markers was found in adults, while there was only brief activation of these markers in pediatric NE cases. The small number of control patient samples could be a limitation of this study. Therefore, more studies using larger control cohorts could further advance our understanding of the age-related differences in NE pathogenesis.

## 4. Materials and Methods

### 4.1. Subjects

Serum and urine samples were collected from 11 pediatric (11 ± 4.5 years old) and 129 adult (39.6 ± 13.6 years old) NE patients during an outbreak in 2019. There were six cases from an outbreak in a children’s summer camp and five cases from two familial NE outbreaks. In addition, serum and urine samples were collected from 11 control children (11.9 ± 3.5 years old) and 26 control adults (38 ± 16.3 years old). The absence of HFRS history and a negative serology result for orthohantavirus antigens was the criteria for controls. Multiple serum samples were collected from each NE patient during the febrile (6.1 ± 0.9 days; 6 children, 12.8 ± 2.1 years old and 23 adults, 36.6 ± 11.7 years old), the oliguric (10.5 ± 2.0 days; 6 children, 10.8 ± 4.4 years old and 66 adults, 40.5 ± 14.3 years old), and polyuric (18.1 ± 2.0 days; 11 children 10.3 ± 4.4 years old and 109 adults, 39.3 ± 14.2 years old) phases. Clinical records and laboratory results were also collated for these patients. The diagnosis of NE was established based on clinical presentation and was serologically confirmed by the detection of anti-orthohantavirus antibodies. Samples were collected following the standard operating procedure protocol in the hospital for the diagnosis of orthohantavirus infection, and aliquots were stored at −80 °C until used.

The severity of the disease was determined according to the national diagnostic criteria for infectious disease by Yuschuk and Vengerov [59,60] as well as recently updated criteria [60] as mild, moderate and severe forms. The mild form was characterized by fever (<38.0 °C), oliguria (<900 mL/day), microproteinuria, microhematuria, blood urea nitrogen (BUN) levels within the norm and increased sCr levels (<130 µM/L). Symptoms of moderate NE were fever (39.5 °C), headache, frequent vomiting, lumbar pain, abdominal pain, hemorrhages, oliguria (<300 mL/day), increased serum levels of BUN (>18 mM/L) and sCr (>300 µM/L). The severe form was characterized by shock, symptoms of hemorrhage, oliguria (<300 mL/day) or anuria, increased serum levels of BUN (>18.5 mM/L) and sCr (>300 µM/L).

Febrile, oliguric and polyuric phases were defined according to Yuschuk and Vengerov [60]. The febrile phase was characterized by fever (>37 °C), with headache, nausea and myalgia often described in this phase. Some patients had vision problems, eye pain, hyperemia of the face and chest, petechia and nasal bleeding. The duration of this phase was between 3 and 10 days. In the oliguric phase, patient conditions worsened, and they presented with symptoms of AKI and hemorrhages. The main symptom defining this phase was oliguria (<900 mL/day urine output). This phase lasted between 6 and 14 days. The polyuric phase (9–13 days of the disease) was mainly characterized by polyuria (>2500 mL/day urine output).

### 4.2. Ethics Statement

The Ethics Committee of the Kazan State Medical Academy (KSMA; protocol 4/09 of the KSMA Ethics Committee meeting dated 26 September 2019) and Kazan Federal University approved this study (article 20, Federal Law “Protection of Health Right of Citizens of Russian Federation” N323-FZ, 21 November 2011). Signed informed consent was obtained from each patient and controls according to the guidelines adopted under these protocols. The informed consent for minors was signed by the guardian.

### 4.3. Othohantavirus Enzyme-Linked Immunosorbent Assay (ELISA)

The Hantagnost diagnostic ELISA kit (Institute of Poliomyelitis and Viral Encephalitis, Moscow, Russia) was used to detect hantavirus-specific antibodies as per the manufacturer’s instructions [61]. This ELISA kit is based on using Hantaan, Seoul и Dobrava/Belgrad and Puumala antigens and has specificity and sensitivity 100% (96.6–100%) and 100% (96.8–100%), respectively. Briefly, NE patient and control serum was diluted 1:100 (PBS) and incubated for 60 min at 37 °C in a 96-well plate with pre-adsorbed orthohantavirus antigens. Following washes (3×; 0.5% Tween20 in PBS, PBS-T), wells were incubated with anti-human-IgG-HRP conjugated antibodies (1:10,000 in PBS-T, Amerixan Qualex Technologies, San Clemente, California, USA) for 30 min at 37 °C. Then, following washes (3×; 0.5% Tween20 in PBS), wells were incubated with 3,3’,5,5’ Tetramethylbenzidine (Chema Medica, Moscow, Russia). The reaction was stopped by adding 10% phosphoric acid (TatKhimProduct, Kazan, Russia). Data were measured using a microplate reader Tecan 200 (Tecan, Switzerland) at OD450 with reference OD650. OD450 values higher than 0.5 were considered positive results.

### 4.4. RT-PCR Detection of PUUV Genome

Total RNA was extracted from 100 μL of blood samples using TRIzol^®^ Reagent (Invitrogen Life Technologies, Waltham, MA, USA), following the manufacturer’s recommendations. cDNA synthesis was performed using Thermo Scientific RevertAid Reverse Transcriptase (Thermo Fisher Scientific, Waltham, MA, USA) according to the manufacturer’s instructions. Nested PCR amplification was undertaken using TaqPol polymerase (Evrogen, Moscow, Russia) following the manufacturer’s instructions. Primer sequences for PCR amplification of the Puumala virus (PUUV) orthohantavirus genome S segment can be found in our previous work [2]. PCR products were cleaned using Isolate II PCR and Gel Kit (Bioline, London, UK), in accordance with the manufacturer’s manual. PCR products were sequenced using ABI PRISM 310 Big Dye Terminator 3.1 sequencing kit (ABI, Waltham, MA, USA) as per the manufacturer’s specifications. The obtained segment nucleotide sequences were analyzed and uploaded online for publication in the GenBank database, under accession numbers summarized in Appendix A.

### 4.5. Phylogenetic Analysis

Multiple nucleotide sequences of the PUUV strains were first aligned using the MegAlign program (Clustal W algorithm) from the DNASTAR software v. 7.1.0 package Lasergene (DNASTAR, Madison, WI, USA; accessed on 12 February 2022 [62], and then used for phylogenetic analysis with MEGA v6.0 [63]. The Maximum Likelihood (ML) method using the Tamura–Nei model incorporated in Mega v6.0. was set to infer the phylogenetic tree. Bootstrap values less than 70% were excluded from the tree shown. Reference nucleotide sequences of PUUV strains from bank voles and HFRS patients were downloaded from the GenBank, and are summarized in Appendix A [38,41,42]. The sequence of *Tula orthohantavirus* of the accession number AF164093, S segment was used as an outgroup.

### 4.6. Multiplex Analysis

Serum levels of 48 analytes were analyzed using Bio-Plex (Bio-Rad, Hercules, CA, USA) multiplex magnetic bead-based antibody detection kits following the manufacturer’s instructions. Human kidney toxicity panel 1 (Bio-Rad, Hercules, CA, USA) was used to analyze urine samples according to the manufacturer’s recommendations. Kidney toxicity panel 1 detects urine levels of calbindin, clusterin, glutathione S-transferase-π (GST-π), IL-18, kidney injury molecule-1 (KIM-1) and monocyte chemotactic protein-1 (MCP-1; CCL2). Multiplex kits, Bio-Plex Pro Human Cytokine 48 panel was used in this study. Serum aliquots (50 μL) were analyzed where a minimum of 50 beads per analyte were acquired. Median fluorescence intensities were collected using a Luminex 100 or 200 analyzer (Luminex, Austin, TX, USA). Each sample was analyzed in triplicate. Data collected was analyzed with MasterPlex CT control software v.3 and MasterPlex QT analysis software v.3 (MiraiBio, San Bruno, CA, USA). Standard curves for each cytokine were generated using standards provided by the manufacturer. Data were analyzed using MasterPlex CT control software v.3 and MasterPlex QT analysis software v3 (MiraiBio, Alameda, CA, USA).

### 4.7. Statistical Analysis

Statistical analysis was performed in the R environment (R-project, 2020, https://www.r-project.org/, accessed on 1 November 2020). The data were not normally distributed (as assessed by the Shapiro–Wilk normality test). Statistically significant differences between comparison groups were accepted as *p* < 0.05, assessed by the Kruskal–Wallis test with Benjamini–Hochberg (BH) adjustment for multiple comparisons.

## Figures and Tables

**Figure 1 ijms-24-07016-f001:**
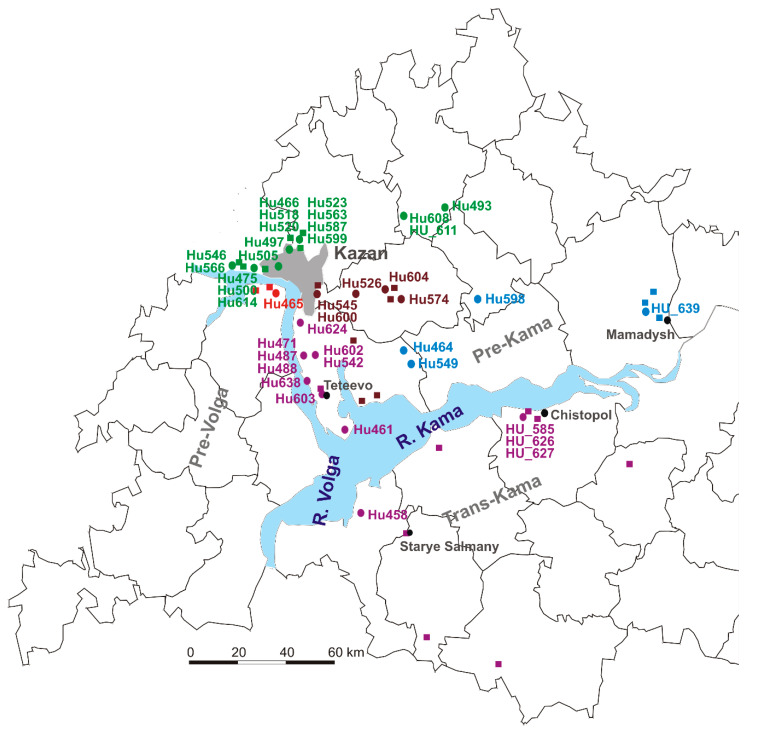
The sites of *orthohantavirus* exposure. The green circles and names (PUUV strains Hu466, Hu475, Hu493, Hu497, Hu500, Hu505, Hu518, Hu520, Hu523, Hu546, Hu563, Hu566, Hu587, Hu599, Hu608, HU_611 and Hu614 from NE patients). The brown circles (Hu526, Hu545, Hu574, Hu 600 and Hu604 strains were closely related to PUUV strains from habitats of bank voles in the east and southeast suburbs of Kazan). The blue circles and names (Hu464, Hu549, Hu598 and HU_639 strains were closely related to PUUV strains from Mamadysh in the east of the Pre-Kama area). The red circles and names for (Hu465). The purple circles (PUUV HU_585, HU_626 and HU_627, Hu461, Hu471, Hu487, Hu488, Hu542, Hu578, Hu602, Hu603, Hu638 and Hu 624 strains).

**Figure 2 ijms-24-07016-f002:**
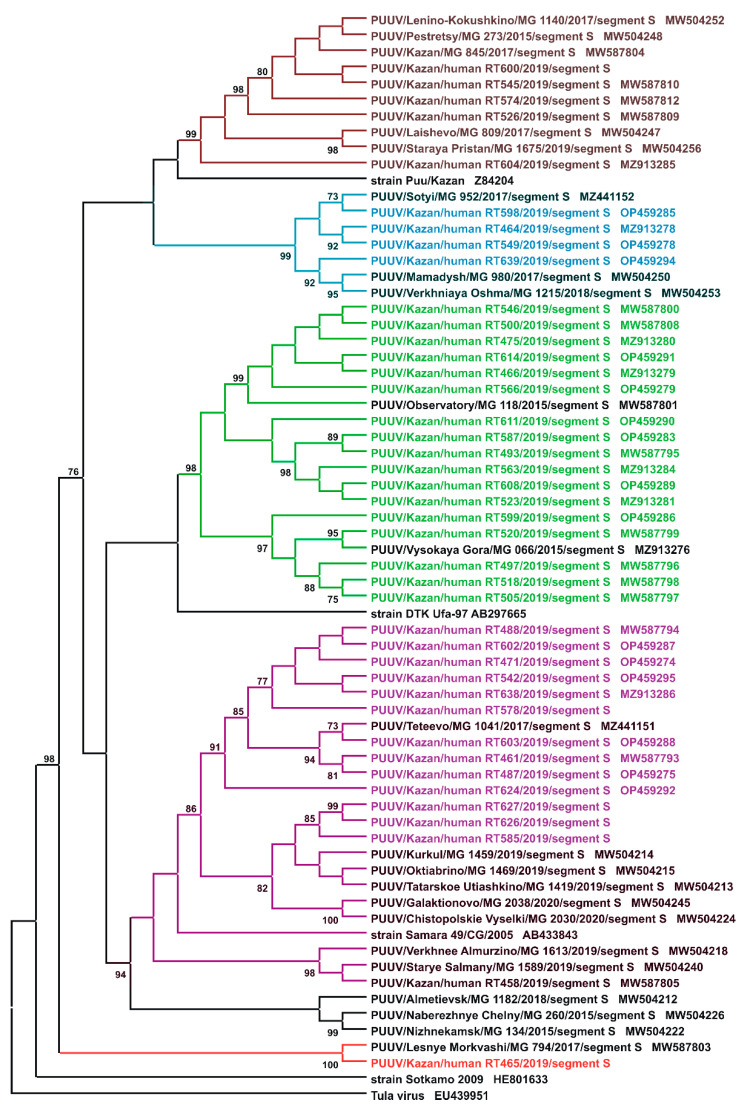
Phylogenetic tree. Sequencing and phylogenetic analysis of PUUV strains from humans and rodents.

**Figure 3 ijms-24-07016-f003:**
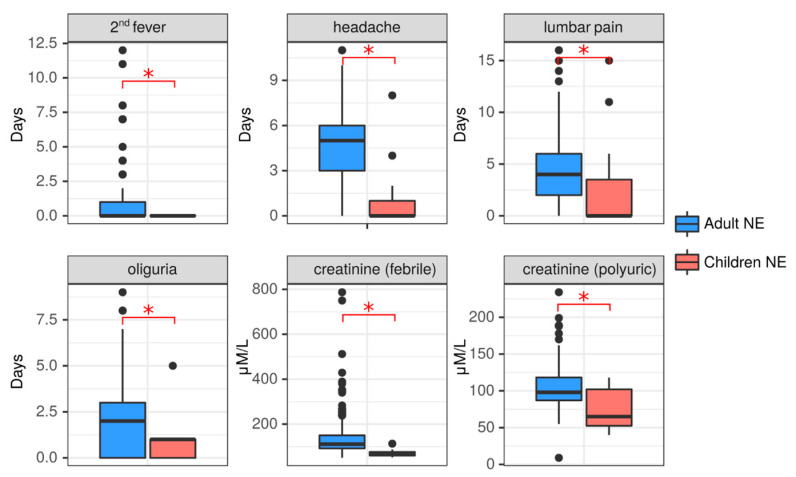
Clinical comparison of NE in adults and children. Duration of the symptoms 2nd fever, headache, lumbar pain and oliguria were recorded in both adult (blue) and child (red) NE patients. Serum creatine levels at the febrile and polyuric phases were also compared between adults (blue) and children (red) with NE. Data is presented as boxplotwith asterisks denoting statistical significance between adults and children as determined by Kruskal-Wallis test (*p* < 0.05).

**Figure 4 ijms-24-07016-f004:**
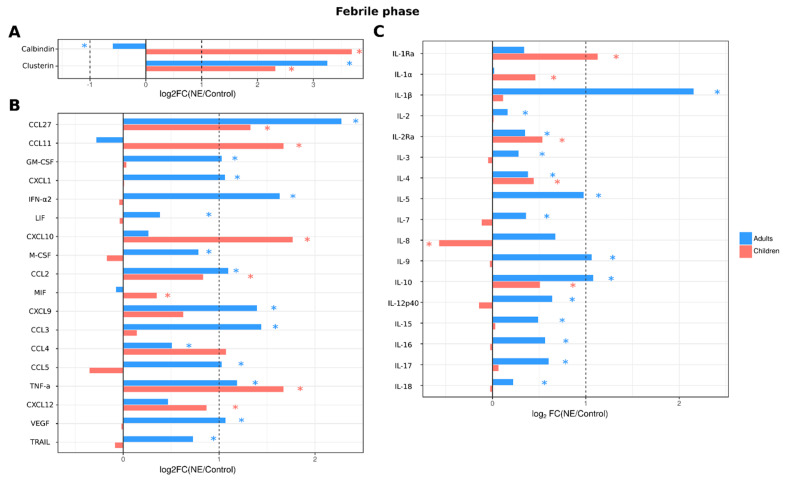
Analysis of serum cytokine and kidney toxicity markers in adults and children diagnosed with NE in the febrile phase. The Bio-Plex (Bio-Rad, Hercules, CA, USA) multiplex magnetic bead-based antibody detection kit was used to measure (**A**) markers of kidney toxicity in the urine; (**В**) serum chemokines or (**С**) serum interleukins in adults (*n* = 129) and children (*n* = 11). Data are presented as Log2 fold change in comparison to non-NE control samples. Asterisks denote statistical significance determined by Kruskal-Wallis test with BH adjustment (*p* < 0.05). Dotted line indicates fold change of 1.

**Figure 5 ijms-24-07016-f005:**
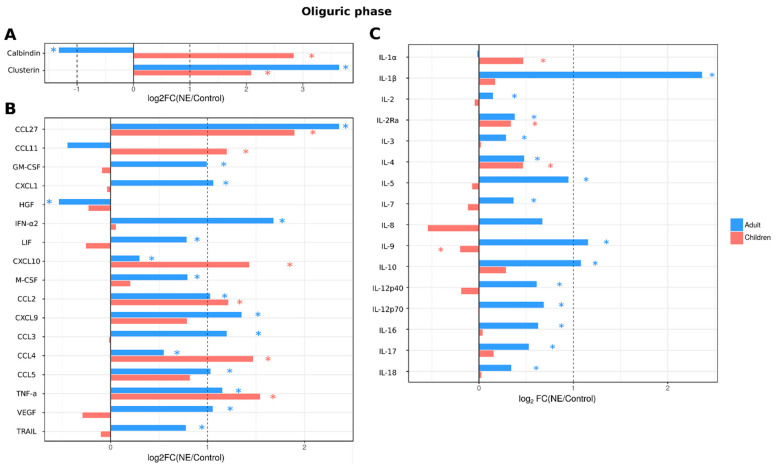
Analysis of serum cytokine and kidney toxicity markers in adults and children diagnosed with NE in oliguric phase. The Bio-Plex (Bio-Rad, Hercules, CA, USA) multiplex magnetic bead-based antibody detection kit was used to measure (**A**) markers of kidney toxicity in the urine; (**В**) serum chemokines or (**С**) serum interleukins in adults (*n* = 129) and children (*n* = 11). Data are presented as Log2 fold change in comparison to non-NE control samples. Asterisks denote statistical significance determined by Kruskal-Wallis test with BH adjustment (*p* < 0.05). Dotted line indicates fold change of 1.

**Figure 6 ijms-24-07016-f006:**
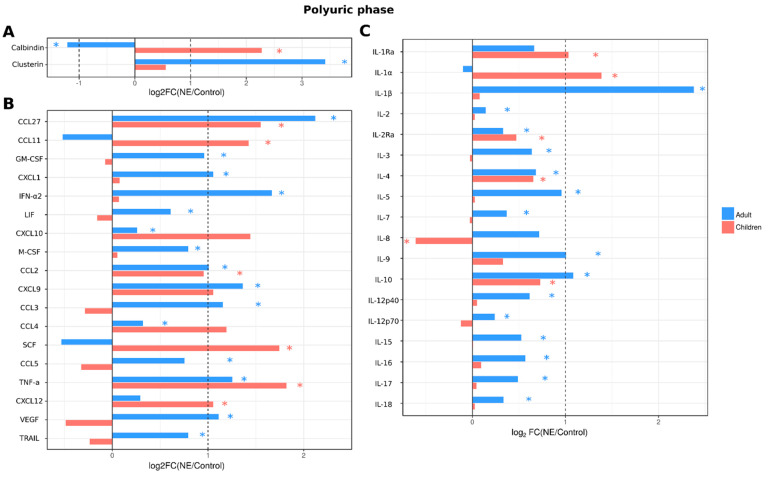
Analysis of serum cytokine and kidney toxicity markers in adults and children diagnosed with NE in polyuric phase. The Bio-Plex (Bio-Rad, Hercules, CA, USA) multiplex magnetic bead-based antibody detection kit was used to measure (**A**) markers of kidney toxicity in the urine; (**В**) serum chemokines or (**С**) serum interleukins in adults (*n* = 129) and children (*n* = 11). Data are presented as Log2 fold change in comparison to non-NE control samples. Asterisks denote statistical significance determined by Kruskal-Wallis test with BH adjustment (*p* < 0.05). Dotted line indicates fold change of 1.

**Table 1 ijms-24-07016-t001:** Clinical signs and symptoms in adult NE patients.

Sign/Symptom	%
Fever	129/129 (100%)
2nd fever	37/129 (28.68%)
Headache	103/129 (79.84%)
Lumbar pain	110/129 (85.27%)
Oliguria	92/129 (71.32%)
Anuria	5/129 (3.88%)
Abdominal pain	49/129 (37.98%)
Nausea	53/129 (41.09%)
Vomiting	38/129 (29.46%)
Diarrhea	43/129 (33.33%)
Cough	19/129 (14.73%)
Hemorrhages	9/129 (6.98%)
Fog in eyes	49/129 (37.98%)

**Table 2 ijms-24-07016-t002:** Clinical laboratory testing results in adult NE patients and controls.

Analyte	Phase	Adult NE (Mean ± SD)	Adult Control (Mean ± SD)	*p* Value
Urea, mM/L	Febrile	8.39 ± 6.03	4.29 ± 0.35	<0.0001 *
	Oliguric	10.13 ± 7.50		<0.0001 *
	Polyuric	6.44 ± 8.87		0.004 *
sCr, µM/L	Febrile	148.85 ± 112.36	78.84 ± 10.50	<0.0001 *
	Oliguric	172.64 ± 141.60		<0.0001 *
	Polyuric	103.60 ± 31.93		<0.0001 *
K, mEq/L	Febrile	4.04 ± 0.43	3.51 ± 0.94	<0.0001 *
	Oliguric	5.43 ± 11.91		<0.0001 *
Thrombocytes, cells/μL	Febrile	90.71 ± 56.30	216.31 ± 11.92	<0.0001 *
	Oliguric	143.37 ± 82.75		<0.0001 *
	Polyuric	256.71 ± 83.75		0.068

* the Maximum Error Probability Criterion.

**Table 3 ijms-24-07016-t003:** Clinical signs and symptoms in pediatric NE patients.

Sign/Symptom	Occurrence (%)
Fever	11/11 (100%)
2nd fever	0/11 (0%)
Headache	3/11 (27.27%)
Lumbar pain	4/11 (36.36%)
Oliguria	7/11 (63.64%)
Anuria	0/11 (0%)
Abdominal pain	5/11 (45.45%)
Nausea	5/11 (54.55%)
Vomiting	5/11 (45.45%)
Diarrhea	1/11 (9.09%)
Cough	0/11 (0%)
Hemorrhages	1/11 (9.09%)
Fog in eyes	3/11 (27.27%)

**Table 4 ijms-24-07016-t004:** Clinical laboratory testing results in pediatric NE patients and controls.

Analyte	Phase	Children NE (Mean ± SD)	Children Control (Mean ± SD)	*p* Value
Urea, mM/L	Febrile	5.18 ± 1.99	2.99 ± 0.51	<0.0001 *
	Oliguric	10.29 ± 9.95		0.001 *
	Polyuric	5.28 ± 1.11		0.0002 *
sCr, µM/L	Febrile	70.55 ± 17.33	43.52 ± 8.72	0.21
	Oliguric	144.64 ± 118.82		0.003 *
	Polyuric	75.18 ± 28.85		0.018 *
K, mEq/L	Febrile	4.06 ± 0.28	4.19 ± 0.50	0.92
	Oliguric	4.40 ± 0.50		0.43
Thrombocytes, cells/μL	Febrile	91.73 ± 51.39	235.73 ± 44.10	0.0001 *
	Oliguric	150.45 ± 69.70		0.03 *
	Polyuric	266.55 ± 68.31		0.57

* the Maximum Error Probability Criterion.

## Data Availability

The original contributions presented in the study are included in the article/Appendix A and available in NCBI database, Further inquiries can be directed to the corresponding author.

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
