# Peer review of "Differential Cytokine Responses and the Clinical Severity of Adult and Pediatric Nephropathia Epidemica"

_ijms, 2023, doi:10.3390/ijms24087016_

Round 1
Reviewer 1 Report
1. "the oliguric (10.5±2.0 days; 6 children, 10.8±4.4 years old and in 66 adults, 40.5±14.3 years old) and polyuric (18.1±2.0 days; 11 children 10.3±4.4 years old and 109 adults, 39.3±14.2 years old) phases. Clinical records and laboratory results were also collated for these patients. The diagnosis of NE was established based on clinical presentation and was" on line 108 and 109 are in a different font compared to the manuscript.
2. We too, have demonstrated an increased level of urinary clusterin in NE as an indicator of kidney injury (Martynova, Maksudova et al. 2018) on line 81 doesn't give the correct notion. In the previous sentence, there is no mention of clusterin. The "We too," part needs to be looked into.
3. Table 2 and 4 contains values which do not fall under the scope defined or required by the Kruskal-Wallis test. Whether it is the Mean with the deviation or the median with the 25th and 75th IQR should be defined. Also, since it is not ranked (assuming how the data is presented), the Statistical methods (2.7) described on line 192 and ahead which mention tests used to calculate the p-value should be written encompassing all the tests used throughout the manuscript, if not specifically. Tests for obvious analysis, for instance, proportion of the total, mentioned under Table 1 and Table 3 need not be mentioned. The authors could also mention that the data was not normally distributed (if so) and why Kruskal-Wallis was preferred over ANOVA. This could be through KS test, SW test, or others.
In my opinion, the authors have analyzed appropriately having checked normality prior to analysis, however, this can be mentioned as well. The manuscript is very well written for readability. Minor edits are required as mentioned. This will be an amazing addition to the literature.
Author Response
- "the oliguric (10.5±2.0 days; 6 children, 10.8±4.4 years old and in 66 adults, 40.5±14.3 years old) and polyuric (18.1±2.0 days; 11 children 10.3±4.4 years old and 109 adults, 39.3±14.2 years old) phases. Clinical records and laboratory results were also collated for these patients. The diagnosis of NE was established based on clinical presentation and was" on line 108 and 109 are in a different font compared to the manuscript.
Agree: font was adjusted: lines 115-118
- We too, have demonstrated an increased level of urinary clusterin in NE as an indicator of kidney injury (Martynova, Maksudova et al. 2018) on line 81 doesn't give the correct notion. In the previous sentence, there is no mention of clusterin. The "We too," part needs to be looked into.
Answer: the manuscript Martynova, Maksudova 2018 is titled as “Urinary Clusterin Is Upregulated in Nephropathia Epidemica” (https://www.ncbi.nlm.nih.gov/pmc/articles/PMC5846347/). Therefore, our statement that we too studied clusterin levels in NE is supported in the manuscript cited. We added DOI number into references to make easy finding the cited manuscripts.
Table 2 and 4 contains values which do not fall under the scope defined or required by the Kruskal-Wallis test. Whether it is the Mean with the deviation or the median with the 25th and 75th IQR should be defined.
Agree: We made changes in the manuscript to address this comment. In Tables 2 and 4 we stated that it is “Mean with the standard deviation”.
Also, since it is not ranked (assuming how the data is presented), the Statistical methods (2.7) described on line 192 and ahead which mention tests used to calculate the p-value should be written encompassing all the tests used throughout the manuscript, if not specifically.
Answer: The corresponding p values were calculated using the Kruskal-Wallis test, because values of clinical features had not a normal distribution. We understand that the Kruskal-Wallis test does not compare mean values, however we have inserted the mean±SD for descriptive purposes.
Tests for obvious analysis, for instance, proportion of the total, mentioned under Table 1 and Table 3 need not be mentioned. The authors could also mention that the data was not normally distributed (if so) and why Kruskal-Wallis was preferred over ANOVA. This could be through KS test, SW test, or others.
Answer: Initially, we completed the Shapiro-Wilk normality test for all studied parameters. Our data appeared to not normally distributed. Therefore, we used the Kruskal-Wallis test for data analysis. We added this statement in M&M section 2.7, lines: 212-213: “The data were not normally distributed (assessed by Shapiro-Wilk normality test).
In my opinion, the authors have analyzed appropriately having checked normality prior to analysis, however, this can be mentioned as well.
Agree: data was checked for normality using Shapiro-Wilk normality test. Statement is added in lines 212-213, M&M section 2.7.
The manuscript is very well written for readability. Minor edits are required as mentioned. This will be an amazing addition to the literature.
Reviewer 2 Report
In this study the authors aimed to determine whether the severity of NE differs in children compared to adult patients. They showed that signs and symptoms of pediatric NE appear to be milder than in adults and that clinical symptoms and signs of AKI were also significantly less pronounced in children.
- The topic is important in the context of current scientific literature.
- Methods and statistics are appropriate.
- Results are relevant.
- Figures and tables are very well detailed.
- Discussion is very well written.
- I support this paper for publication in the current format.
Author Response
In this study the authors aimed to determine whether the severity of NE differs in children compared to adult patients. They showed that signs and symptoms of pediatric NE appear to be milder than in adults and that clinical symptoms and signs of AKI were also significantly less pronounced in children.
- The topic is important in the context of current scientific literature.
- Methods and statistics are appropriate.
- Results are relevant.
- Figures and tables are very well detailed.
- Discussion is very well written.
- I support this paper for publication in the current format.
Answer: we would like to thank you the Reviewer for reviewing this manuscript.
Reviewer 3 Report
The authors have demonstrated substantial differences in clinical and laboratory data in paediatric and adult NE patients. Data provide evidence of a milder form of NE in children than adults. They suggest that variations in cytokine activation could explain the differences in clinical presentation. Cytokines associated with activation of Th1 lymphocytes are prominent in adults, while they were absent in paediatric NE patient serum.
The topic is original and relevant in the field, because it addresses a specific gap in the field.
The conclusions are consistent with the evidence presented. Definitely, the authors address the main question posed.
The bibliographical references are corrent and cover a wide range of articles that support the research developed.
The Tables and Figures contain the correct scientific information with the necessary statiitical parameters to give them credibility.
The article deserves to be published because it meets all the necessary standards to be a reference for scientists working in this therapeutic area.
Author Response
The authors have demonstrated substantial differences in clinical and laboratory data in paediatric and adult NE patients. Data provide evidence of a milder form of NE in children than adults. They suggest that variations in cytokine activation could explain the differences in clinical presentation. Cytokines associated with activation of Th1 lymphocytes are prominent in adults, while they were absent in paediatric NE patient serum.
The topic is original and relevant in the field, because it addresses a specific gap in the field.
The conclusions are consistent with the evidence presented. Definitely, the authors address the main question posed.
The bibliographical references are corrent and cover a wide range of articles that support the research developed.
The Tables and Figures contain the correct scientific information with the necessary statiitical parameters to give them credibility.
The article deserves to be published because it meets all the necessary standards to be a reference for scientists working in this therapeutic area.
Answer: we would like to thank you the reviewer for reviewing this manuscript.
Reviewer 4 Report
The study aimed to determine the prevalence of orthohantavirus-specific antibodies in serum and urine samples of patients with nephropathia epidemica (NE) during an outbreak in 2019 in Russia. The study included 11 children and 129 adults with NE, and 11 control children and 26 control adults. Samples were collected during different phases of the disease, and severity was classified based on the National Diagnostic Criteria for infectious diseases by Yuschuk and Vengerov. The study followed standard operating procedures, and informed consent was obtained from all participants. The Hantagnost diagnostic ELISA kit was used to detect orthohantavirus-specific antibodies.
In sum, the study presented in this article provides valuable insights into the differential cytokine responses and clinical severity of adult and paediatric Nephropathia Epidemica (NE). The use of a kidney toxicity panel to analyze urine samples from patients is a notable addition to the study, and the authors' conclusions are well-supported by the data presented. The study also highlights the need for further research into NE pathogenesis in children, given the limited number of paediatric NE cases studied to date.
This study adds to the growing body of literature on the differential clinical presentation of NE in adults and children. The authors' analysis of serum cytokines and kidney toxicity markers in patients provides a valuable contribution to the field, and their findings suggest that differences in cytokine activation may underlie the observed age-related differences in NE severity. However, it should be noted that the small sample size of paediatric NE cases included in the study may limit the generalizability of the findings.
In this study, we sought to determine whether the severity of NE differs in children 89 compared to adult patients. We show that signs and symptoms of paediatric NE appear 90 to be milder than in adults and that clinical symptoms and signs of AKI were also signif-91 icantly less pronounced in children. A fewer number of cytokines were found to be acti-92 vated in sera from children, suggesting that the “cytokine storm” contributes more to 93 pathogenesis in older patients. Additionally, prominent activation of cytokines associated 94 with activation of Th1 lymphocytes, and a prolonged activation of kidney injury markers 95 was found in adults, while they were only briefly activated in children.
Comment: this section should be rewritten
I have some major comments:
The study provides a clear description of the study design and methodology. The inclusion of a control group strengthens the study's validity and reduces the potential for bias. However, the sample size of the control group is relatively small, which may limit the generalizability of the results.
The use of the Hantagnost diagnostic ELISA kit to detect orthohantavirus-specific antibodies is an established method, but it is worth noting that the sensitivity and specificity of the kit may vary depending on the virus strain and the patient population. Additionally, the study did not perform confirmatory testing, such as western blotting or plaque reduction neutralization testing, which may have increased the accuracy of the results.
The study's classification of disease severity based on the National Diagnostic Criteria for infectious diseases by Yuschuk and Vengerov may not be widely accepted, as the criteria were developed in 2020 and may not have been validated in other populations. Furthermore, the clinical presentation and severity of the disease may vary depending on the patient population, and the study did not control for confounding variables that may have affected the results.
Overall, the study provides valuable information on the prevalence of orthohantavirus-specific antibodies in patients with NE during an outbreak in Russia. However, the study's limitations, such as the small sample size of the control group and the lack of confirmatory testing, should be taken into account when interpreting the results.
Author Response
The study aimed to determine the prevalence of orthohantavirus-specific antibodies in serum and urine samples of patients with nephropathia epidemica (NE) during an outbreak in 2019 in Russia. The study included 11 children and 129 adults with NE, and 11 control children and 26 control adults. Samples were collected during different phases of the disease, and severity was classified based on the National Diagnostic Criteria for infectious diseases by Yuschuk and Vengerov. The study followed standard operating procedures, and informed consent was obtained from all participants. The Hantagnost diagnostic ELISA kit was used to detect orthohantavirus-specific antibodies.
In sum, the study presented in this article provides valuable insights into the differential cytokine responses and clinical severity of adult and paediatric Nephropathia Epidemica (NE). The use of a kidney toxicity panel to analyze urine samples from patients is a notable addition to the study, and the authors' conclusions are well-supported by the data presented. The study also highlights the need for further research into NE pathogenesis in children, given the limited number of paediatric NE cases studied to date.
This study adds to the growing body of literature on the differential clinical presentation of NE in adults and children. The authors' analysis of serum cytokines and kidney toxicity markers in patients provides a valuable contribution to the field, and their findings suggest that differences in cytokine activation may underlie the observed age-related differences in NE severity. However, it should be noted that the small sample size of paediatric NE cases included in the study may limit the generalizability of the findings.
In this study, we sought to determine whether the severity of NE differs in children 89 compared to adult patients. We show that signs and symptoms of paediatric NE appear 90 to be milder than in adults and that clinical symptoms and signs of AKI were also signif-91 icantly less pronounced in children. A fewer number of cytokines were found to be acti-92 vated in sera from children, suggesting that the “cytokine storm” contributes more to 93 pathogenesis in older patients. Additionally, prominent activation of cytokines associated 94 with activation of Th1 lymphocytes, and a prolonged activation of kidney injury markers 95 was found in adults, while they were only briefly activated in children.
Comment: this section should be rewritten
We thank the reviewer for their comments and for taking the time to review our manuscript. However, we feel that that a simple statement such as this does not provide enough feedback as to how it should be rewritten. We have however gone through this section and tried to adjust and make some improvements and hope that these are agreeable.
I have some major comments:
The study provides a clear description of the study design and methodology. The inclusion of a control group strengthens the study's validity and reduces the potential for bias. However, the sample size of the control group is relatively small, which may limit the generalizability of the results.
Agree: more controls are always appreciated as it adds strength to validity however sample size is often a limitation of studies such as these due to availability and the nature of this kind of infection, particularly in the case of children. We conducted the statistical power analysis which demonstrated the statistical power for all analytes >0.8.We have added further acknowledgement of the limiting factor of the sample size in the discussion section and feel that our conclusions are still well-supported.
The use of the Hantagnost diagnostic ELISA kit to detect orthohantavirus-specific antibodies is an established method, but it is worth noting that the sensitivity and specificity of the kit may vary depending on the virus strain and the patient population. Additionally, the study did not perform confirmatory testing, such as western blotting or plaque reduction neutralization testing, which may have increased the accuracy of the results.
Answer: Hantagnost is clinically validated test to detect anti-hantavirus antibodies in Russia. This test is routinely used in all clinical laboratories to diagnose hantavirus infection. Also, this test was used in several research papers (https://pubmed.ncbi.nlm.nih.gov/12647557/ , https://pubmed.ncbi.nlm.nih.gov/20420534/ , https://www.sciencedirect.com/science/article/pii/S15671348220009220 https://www.frontiersin.org/articles/10.3389/fpubh.2021.620279/full). This test system was used to demonstrate anti-hantavirus antibodies in patients from, Czech republic and several regions of Russia. The Hantagnost is developed using Hantaan, Seoul и Dobrava/Belgrad and Puumala antigens. It was used for detection of antibodies against Puumala, Seoul, Amur and Hantaan. The test system has sensitivity 100% (96.6%-100%) and specificity 100% (96.8%-100%). We added this information to M&M lines: 154-156: This ELISA kit is based on using Hantaan, Seoul и Dobrava/Belgrad and Pu-umala antigens and has specificity and sensitivity 100% (96.6%-100%) and 100% (96.8%-100%), respectively
The study's classification of disease severity based on the National Diagnostic Criteria for infectious diseases by Yuschuk and Vengerov may not be widely accepted, as the criteria were developed in 2020 and may not have been validated in other populations.
Answer: The National Diagnostic Criteria were developed including the specifics of hantavirus infection in Russia. These criteria are more inclusive as there are two forms of hantavirus infection: severe, also known as hemorrhagic fever with renal syndrome and mild, referred as nephropathia epidemica. We cited the latest edition of the National Diagnostic Criteria, published in 2022. To address this comment we added the older editions (2009 and 2019) into the reference list.
Furthermore, the clinical presentation and severity of the disease may vary depending on the patient population, and the study did not control for confounding variables that may have affected the results. We acknowledge that the control group is small (lines 504-507). Having looked through the literature, the control size used is not uncommon. For example, in study by Tian et al the patient/control ratio was 149/30 (https://www.ncbi.nlm.nih.gov/pmc/articles/PMC9087930/). In another study, done by Dong et al, this ratio was 104/20 (https://link.springer.com/article/10.1007/s11739-015-1195-7).
Unfortunately as mentioned above, sample size is often a limiting factor in studies of this nature, given the route of infection and infection in children is particularly difficult to catch and obtain samples. All samples however were age-matched with controls and control samples were taken from healthy individuals with no history of HFRS and with negative serology test. We believe that these criteria strengthen these control samples and provide a reliable comparison in this study
Overall, the study provides valuable information on the prevalence of orthohantavirus-specific antibodies in patients with NE during an outbreak in Russia. However, the study's limitations, such as the small sample size of the control group and the lack of confirmatory testing, should be taken into account when interpreting the results.
Agree: we made changes in the text (lines 504-507).